# Can floral nectars reduce transmission of *Leishmania*?

**Evan C. Palmer-Young** [1]*, **Ryan S. Schwarz** [2], **Yanping Chen**[1], **Jay D. Evans**[1]

1 USDA-ARS Bee Research Laboratory, Beltsville, Maryland, United States of America, 2 Department of Biology, Fort Lewis College, Durango, Colorado, United States of America

* ecp52@cornell.edu, evan.palmer-young@usda.gov

## Abstract

### Background

Insect-vectored *Leishmania* are responsible for loss of more disability-adjusted life years than any parasite besides malaria. Elucidation of the environmental factors that affect parasite transmission by vectors is essential to develop sustainable methods of parasite control that do not have off-target effects on beneficial insects or environmental health. Many phytochemicals that inhibit growth of sand fly-vectored *Leishmania*—which have been exhaustively studied in the search for phytochemical-based drugs—are abundant in nectars, which provide sugar-based meals to infected sand flies.

### Principle findings

In a quantitative meta-analysis, we compare inhibitory phytochemical concentrations for *Leishmania* to concentrations present in floral nectar and pollen. We show that nectar concentrations of several flowering plant species exceed those that inhibit growth of *Leishmania* cell cultures, suggesting an unexplored, landscape ecology-based approach to reduce *Leishmania* transmission.

### Significance

If nectar compounds are as effective against parasites in the sand fly gut as predicted from experiments *in vitro*, strategic planting of antiparasitic phytochemical-rich floral resources or phytochemically enriched baits could reduce *Leishmania* loads in vectors. Such interventions could provide an environmentally friendly complement to existing means of disease control.

**Data Availability Statement:** All relevant data are within the manuscript and its Supporting information files.

**Funding:** This project was supported by the USDA Agricultural Research Service Beltsville Bee

## Author summary

*Leishmania* parasites infect over a million people each year—including over 200,000 infections with deadly visceral leishmaniasis—resulting in a greater health burden than any human parasite besides malaria. *Leishmania* infections of humans are transmitted by blood-feeding sand flies, which also consume floral nectar. Nectar contains many

Research Laboratory in house fund; USDA-NIFA Pollinator Health Grant 2020-67013-31861 to JDE and YPC; and a North American Pollinator Protection Campaign Honey Bee Health Improvement Project Grant and an Eva Crane Trust Grant to ECPY and JDE. Funders had no role in study design, data collection and interpretation, or publication.

**Competing interests:** The authors have declared that no competing interests exist.

chemicals that inhibit *Leishmania* growth and are candidate treatments for infection of humans. However, these same compounds could also reduce infection in nectar-consuming sand flies. By combining existing data on the chemistry of nectar and sensitivity of *Leishmania* to plant compounds, we show that some floral nectars contain sufficient chemical concentrations to inhibit growth of insect-stage *Leishmania*. Our results suggest that consumption of these nectars could reduce parasite loads in sand flies and transmission of parasites to new human hosts. In contrast to insecticide-based methods of sand fly control, incorporation of antiparasitic nectar sources into landscapes and domestic settings could benefit public health without threatening beneficial insects. These findings suggest an unexplored, landscape-based approach to reduce transmission of a major neglected tropical disease worldwide.

## Introduction

Plant secondary metabolites have a long history of use against human disease and provide the basis for both traditional medicines and many modern drugs [1], including treatments for neglected tropical diseases [2]. The sand fly-vectored *Leishmania* parasites are estimated to cause disease in >2 million humans each year, with 10% of the world's population at risk, and have a greater health burden (as measured by loss of disability-adjusted life years) than any human parasite besides malaria [3]. These infections include an estimated >0.2M cases of visceral leishmaniasis, which, if untreated, results in >90% patient mortality [3,4]. Due to their clinical significance, *Leishmania* spp. have been studied intensively in a search for affordable and effective treatments for human infections [5], including exhaustive testing of plant extracts and their components against both mammal- and insect-associated parasite life stages [2,6]. These studies have suggested new treatments for trypanosomatid-associated infections of humans [7] and related parasites of beneficial insects [8,9]. As in humans, antimicrobial phytochemicals can enhance resistance to infection in plants themselves [10] and in other plant-consuming animals, including insects [11].

The diets of blood-feeding, disease-vectoring insects such as sand flies and mosquitoes include sugar-containing plant tissues as well as blood [12]. Sugar sources differentially affect not only vector survival, but also the development of parasitic *Plasmodium falciparum* malaria in *Anopheles* mosquitoes [13] and *Leishmania major* in sand flies (*Phlebotomus papatasi*) [14], with effects mediated by secondary metabolites [15,16]. Sand flies feed on plant sugars between acquisition and transmission of *Leishmania* to humans and other mammals [17], as demonstrated by caging flies with dye-infused branches, spectrophotometric detection of sugars or plant cell walls in the gut, and molecular analysis of field-collected flies showing the presence of plant DNA [18–20]. The importance of dietary sugars is evident from their effects on fly longevity. Flies survive less than a week under sugar source-poor desert conditions [21] and less than 2 weeks when reared on comparatively sugar-poor branches [22], but more than 7 weeks on 20% sucrose solution [22]. The abundance of sugar meal-inducible glucosidases expressed by the sand fly and by its *Leishmania* parasites provide additional evidence of mutual adaptation to an omnivorous lifestyle that exploits diverse plant sugars as food sources [23–25], and that sugar sources could be manipulated to control vectors and their parasites [16,22].

Although sand flies may acquire sugar meals from plant sap, fruit, or aphid- or cicada-derived honeydew [26], floral nectar appears to be a preferred food source, as evidenced by the attractiveness of flowering bushes and branches (relative to those soiled with honeydew) in a desert oasis [12]. The small (<1 μL) meal sizes of sand flies [23] would make the concentrated

sugars found in nectar a profitable foraging resource, in spite of the small volumes available at each flower, explaining the general attractiveness of flowering plant food sources to sand flies and related dipterans [27]. The size of sugar meals is, however, impressive on a mass-specific basis—increasing the mass of females by >30% over 48 h [28]—consistent with the strong effects of meal chemistry on gut-dwelling *Leishmania*.

The role of nectar chemistry in insect disease ecology has recently been highlighted by work on infections of pollinators. Floral nectar and pollen, their constituent secondary metabolites, and the composition of flowering plant communities can ameliorate trypanosomatid growth and infection in bumble bees [8,9,29–31]. Both nectar and pollen—which may mix with and influence the chemistry of nectar at flowers [32]—contain diverse secondary metabolites that shape plant-pollinator ecology and plant-microbe ecology [33–37]. Flavonoids are one class of antimicrobial and antileishmanial compounds [38,39] that are ubiquitous in both nectar and pollen, with concentrations in pollen often exceeding 1% of total dry matter [40,41]. This suggests that consumption of secondary metabolite-rich nectars could mitigate *Leishmania* transmission by reducing infection intensity in nectar-feeding sand fly vectors [12], pointing to a new strategy for drug- and insecticide-free disease control. However, despite appreciation for the clinical antileishmanial potential of plant metabolites [2], growing recognition of the role of plant metabolites—including those in nectar and pollen—in insect infection, and the critical role of plant sugars in sand fly diets, there has been surprisingly little investigation into the potential for antileishmanial phytochemicals in the diets of sand flies to mitigate *Leishmania* transmission [14,15].

To assess the potential for floral resource-associated phytochemicals to reduce vector-borne infection, we compared phytochemical concentrations previously shown to inhibit *Leishmania* to concentrations previously found in floral nectar and pollen. Our synthesis of prior work on *Leishmania* phytochemical sensitivity with nectar and pollen secondary chemistry shows that many floral nectars contain antileishmanial compounds at concentrations sufficient to inhibit parasite growth. These findings suggest an unexplored, landscape ecology-based approach to reduce transmission of widespread and virulent *Leishmania* infections. If phytochemical concentrations that inhibit *Leishmania in vitro* are equally effective in the sand fly gut, incorporation of antiparasitic nectar sources into landscapes and domestic settings could simultaneously benefit pollinator and public health.

## Methods

We compared the flavonoid concentrations found in a previous survey of methanolic extracts from 26 floral nectars and 28 pollens [40,42] with previously published results from *in vitro* screening of various *Leishmania* spp. (Table A in S1 Text). We focused on flavonoids because these compounds were the most consistently present class of compounds across both nectar and pollen [40] and—particularly in the case of quercetin—some of the most potent and selective compounds against *Leishmania* [39,43,44]. To prevent overestimation of inhibitory potential that could result from including flavonoids of lesser or unknown antiparasitic activity, we further distinguished between total flavonoid concentrations and those with a kaempferol, quercetin, apigenin, or luteolin aglycone, each of which has well-documented antileishmanial effects [39,45,46] (Table A in S1 Text).

We analyzed micromolar concentrations to enable pooling across compounds with different parent flavonoids and glycosides. Flavonoid glycosides—including those of quercetin and kaempferol—can be less potent against *Leishmania* than are their parent aglycones [39], which can more easily cross cell membranes [47]. However, we included flavonoid glycosides because these compounds are hydrolyzed by intestinal glucosidases—a variety of which are found in

sand flies [28]—to their corresponding aglycones [48,49]. These glucosidases have been shown to form antileishmanial aglycones from glycosylated coumarins in intestinal extracts [16]. We focus our discussion on nectar because sand flies, like other Diptera, do not have chewing mouthparts that would enable direct consumption of pollen and other solid foods [20]. However, incidental presence of pollen in nectar can dramatically increase the nectar's concentrations of amino acids [50], with ecologically relevant effects on nectar-feeding insects [32]. Pollen could similarly affect nectar's phytochemical profile and antimicrobial effects. For example, presumably pollen-derived cinnamic acid-spermidine conjugates were found in nectar of two species in our previous survey—*Digitalis purpurea* and *Helianthus annuus* [40]. In *H. annuus*, nectar concentrations averaged 1.7% of pollen concentrations, despite exclusion of large insects that contribute to such "contamination" [50] for 24 h prior to sampling. We therefore also discuss pollen concentrations that exceed the *Leishmania* IC50 estimates by >100-fold, on the grounds that the much (235-fold [40]) higher flavonoid concentrations found in pollen could meaningfully alter the antiparasitic activity of nectar, even when pollen accounts for <1% of nectar volume.

## Results

We compiled 18 *Leishmania* IC50 estimates for 4 flavonoids—quercetin (n = 8), kaempferol (n = 4), apigenin, and luteolin (n = 3 each) that have been relatively well studied for effects on *Leishmania* spp. cell cultures (Table A in S1 Text). Most (11 of 18) of the assays used the promastigote (i.e., insect-associated) life stage; the remainder used either intracellular (n = 4) or axenic (n = 3) amastigotes (Table A in S1 Text). These *Leishmania* IC50 estimates were then compared to the flavonoid concentrations found in a previous survey of secondary metabolites of nectar and pollen [40].

Flavonoids were found in the nectar of 21 of 26 species (81%) and in pollen of 26 of 28 species (93%), accounting for 30% of the total phytochemical content in nectar and 41% in pollen [40]. Total flavonoid concentrations exceeded 100 µM in 8 of 26 nectars (31%, median concentration 30.9 µM, IQR 4.24–127 µM; median 61.4 µM after exclusion of the five species without nectar flavonoids) and exceeded $10^4$ µM in 18 of 28 pollens (64%, median $1.36 \cdot 10^4$ µM, IQR $4.31 \cdot 10^3$ to $2.30 \cdot 10^4$ µM) (Fig 1). Glycosides of quercetin (found in 9 of 26 nectars and 14 of 28 pollens) and kaempferol (5 of 26 nectars and 19 of 28 pollens) were most common [40].

Compounds with a parent aglycone of quercetin, kaempferol, apigenin, or luteolin accounted for 62% of flavonoid compounds and 54% of molar concentrations in nectar, and 72% of compounds and 75% of molar concentrations in pollen. In nectar, median concentration of this subset of compounds across all species (20.3 µM, IQR 9.37–58.9 µM) was remarkably close to the 23.1 µM median IC50 for *Leishmania* (based on 18 references (Table A in S1 Text)). Concentrations exceeded 100 µM (i.e., more than the highest *Leishmania* IC50 for any of the parent compounds) in nectar from 4 of 26 species (*Dicentra eximia*, *Brassica napus*, *Helianthus annuus*, and *Thymus vulgaris*). In pollen, median concentrations exceeded $10^4$ µM (i.e., >100-fold the greatest *Leishmania* IC50) in pollen from 12 of 28 species, including two species (*Lythrum salicaria* ($1.21 \cdot 10^5$) and *Solidago canadensis* ($1.19 \cdot 10^5$)) with concentrations >$10^5$ µM—over three orders of magnitude above the greatest *Leishmania* IC50 (Fig 1).

Antileishmanial compounds in nectar were not limited to flavonoids. Seven nectars contained chlorogenic acid, with a median concentration (51.2 µM) similar to the IC50 for *L. donovani* promastigotes (54 µM [51]) and 100-fold greater than the IC50 for *L. amazonensis* promastigotes (0.5 µM [52]). The species with the highest median concentration of chlorogenic acid (*Dicentra eximia*, 184 µM) also had the highest concentration of the selected

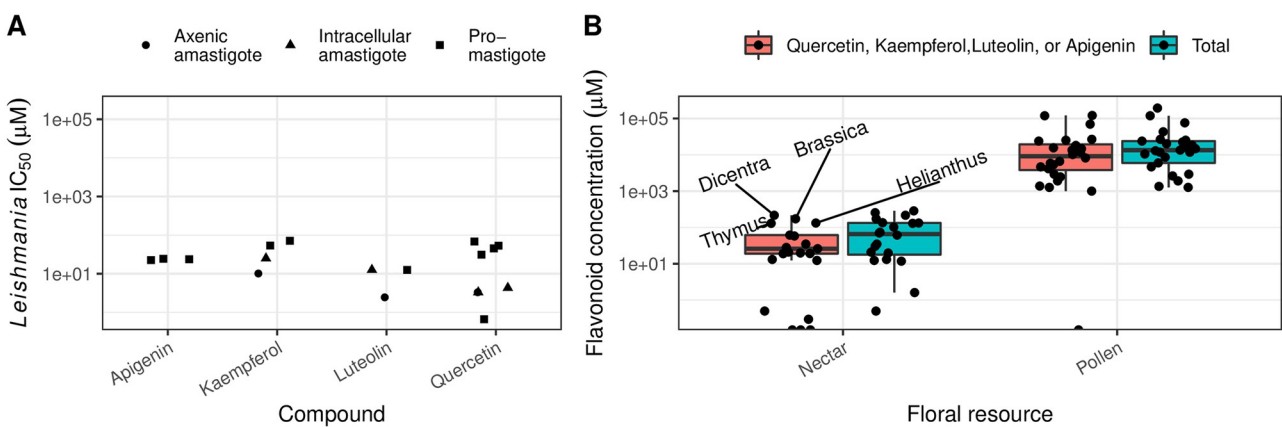

**Fig 1. Published *Leishmania* IC50 estimates for selected flavonoids (A) relative to concentrations of the corresponding compounds in nectar and pollen (B).** Shapes in panel (A) correspond to the *Leishmania* stage tested. Boxplots in panel (B) show medians and interquartile ranges for concentrations of quercetin, kaempferol, apigenin and luteolin derivatives (red boxes) and total flavonoids (blue boxes). Points show median concentrations (pooled across individual samples) by species. Text annotations denote species with >100 µM of the selected flavonoids in nectar (*Brassica napus*, *Dicentra eximia*, *Helianthus annuus*, and *Thymus vulgaris*). Literature references for *Leishmania* IC50 estimates are given in Table A in S1 Text.

flavonoids (Fig 1). Nectar concentrations of two additional species (*Penstemon digitalis*, 134 µM) and *Rhododendron prinophyllum* (56.7 µM) also exceeded the *L. donovani* promastigote IC50 (Fig 2). Chlorogenic acid was also found in seven pollens at up to 3760 µM (*Persea americana*), with a median concentration (1227 µM) over 20-fold greater than that found in nectar and over three orders of magnitude above the *L. amazonensis* promastigote IC50 [52] (Fig 2).

Nectar of one species (*Thymus vulgaris*) contained the caffeic acid-dihydroxyphenyl lactic acid ester rosmarinic acid. Median concentration (165 µM, IQR 87.7–206 µM) was 10-fold greater than the IC50 for *L. donovani* promastigotes (16.3 µM [51])—against which rosmarinic acid and apigenin were the most selective of the compounds evaluated—over 30-fold greater than the 4.8 µM IC50 for *L. amazonensis* amastigotes [52], and over 200-fold greater than the 0.7 µM reported for *L. amazonensis* promastigotes [52] (Fig 3). Nectar of *T. vulgaris* is also notable for its high thymol content (26.1 µg mL⁻¹ [53]), which exceeds six of the eight IC50 values reported for *Leishmania* promastigotes (Table A in S1 Text, [54,55]).

## Discussion

Our synthesis of a previous survey on the quantitative phytochemical composition of nectar and pollen with the extensive body of research on phytochemical-mediated inhibition of Leishmania *in vitro* reveals the potential for floral resources to ameliorate vector-mediated transmission of *Leishmania*. The most common compounds in nectar and pollen—flavonoids and their glycosides—have shown strong inhibitory effects against *Leishmania* [39,44,56]. Our findings indicate that a subset of the floral nectars analyzed to date—including the common garden herb *Thymus vulgaris* (thyme) and the widespread crop species *Helianthus annuus* (cultivated sunflower) contain bioactive flavonoids at concentrations that inhibit growth of diverse *Leishmania in vitro*. Incidentally, both plant species have also been shown to mitigate transmission and infectivity of bumble bee trypanosomatids, including in field mesocosms and landscape surveys [30,31]. Further investigation of the effects of specific nectar and other sugar sources on sand fly infection is needed. However, deliberate encouragement of these and

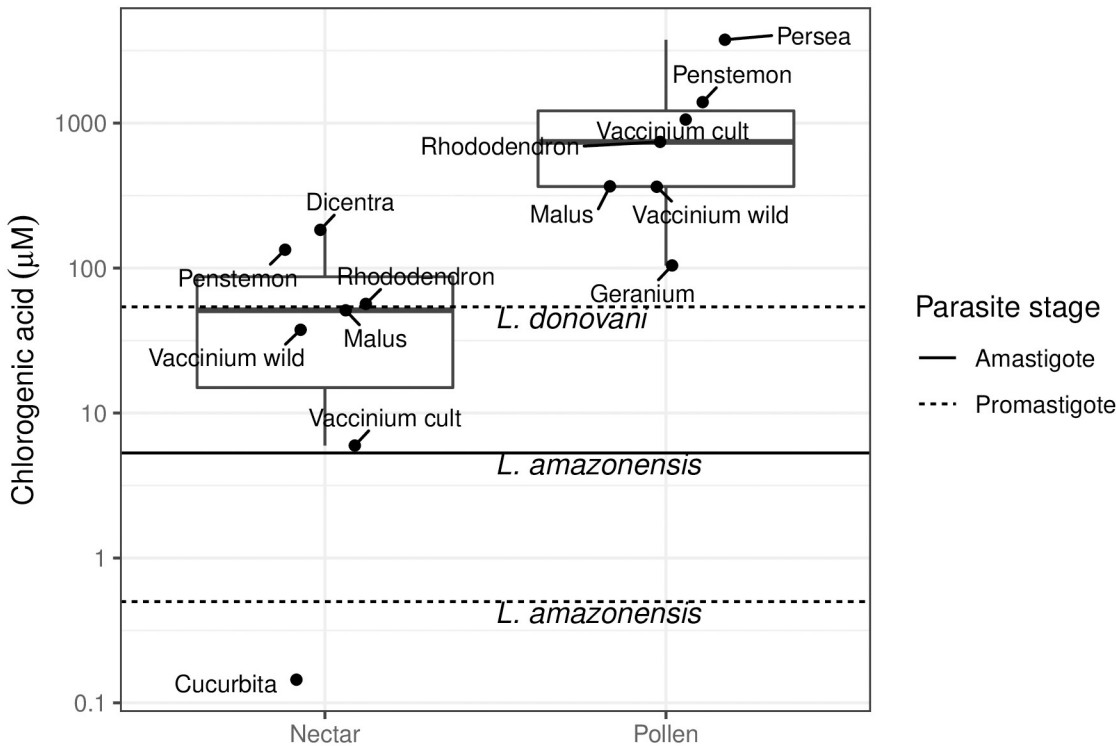

**Fig 2. Concentrations of chlorogenic acid in nectar and pollen in comparison with inhibitory concentrations for *Leishmania*.** Points represent median concentrations from species with detectable chlorogenic acid (sampled in [40]). Horizontal lines show published IC50 values [51,52]. Sampled plant species (labeled by genus) were *Dicentra eximia, Penstemon digitalis, Rhododendron prinophyllum, Malus domestica, Vaccinium corymbosum, Cucurbita pepo, Persea americana,* and *Geranium maculatum.* For *Vaccinium,* "cult" refers to cultivars and "wild" refers to wild plants.

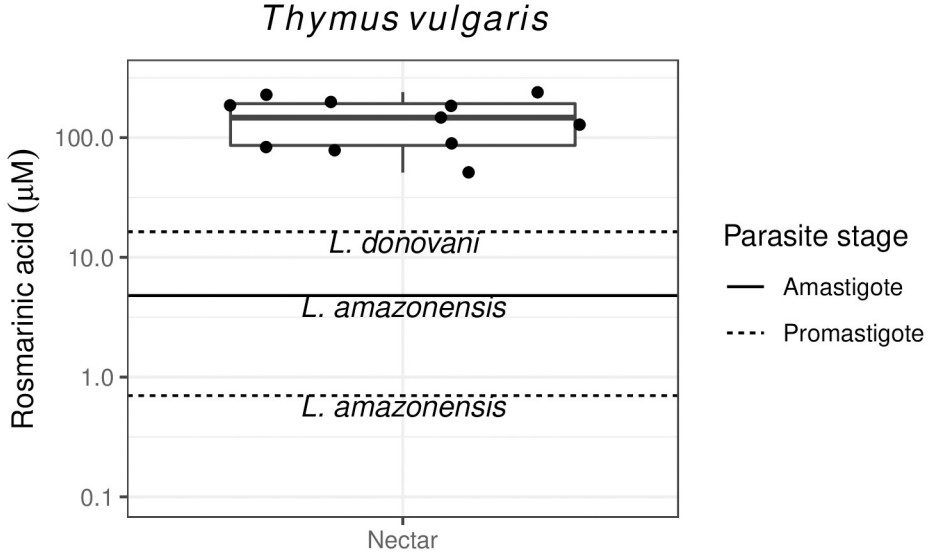

**Fig 3. Concentrations of rosmarinic acid in *Thymus vulgaris* nectar in comparison with inhibitory concentrations for *Leishmania*.** Points represent individual nectar samples (from [40]). Horizontal lines show published IC50 values [51,52].

other plants with trypanosomatid-inhibiting chemistry could reduce infection in disease-vectoring, hematophagous insects as well.

Our data suggest that around 20% of nectars contain flavonoids at strongly antiparasitic concentrations, although this number likely varies by region and season. Our analysis was focused on bee-pollinated species in the Northeast United States, where sand flies are absent, and therefore contained few of the specific plant species naturally used by sand flies in *Leishmania*-endemic regions. However, sand flies have been observed to prefer cultivated gardens (to which such plants could be introduced) over endemic vegetation [20], and have been associated with plants in the same taxonomic families as those represented by the species analyzed here. For example, *Brassica napus* (Brassicaceae) nectar was among the highest in flavonoids; flowers of another member of this family (*Sinapis alba*) elicited feeding by *Phlebotomus papatasi* [57]. Nectar of *Impatiens capensis* (Balsaminaceae) had flavonoid concentrations (20.9 μM, all from strongly antileishmanial compounds) close to the median of the nectars examined (20.3 μM) and the median *Leishmania* IC50 (23.1 μM); branches of the congener *I. balsamina* were fed upon by *Lutzomyia youngi* in Colombian coffee plantations [58]. On the other hand, nectar flavonoid concentrations were considerably lower (1.6 μM total) in *Trifolium pratense*, the only Fabaceae species tested; plants of this family have been strongly associated with sand flies in field sampling [59] and DNA metabarcoding studies [60,61]. Flavonoid concentrations were also low (<1 μM) in *Cucurbita pepo* (Cucurbitaceae) and undetectable in *Catalpa speciosa* (Bignoniaceae), two other plant families associated with sand flies [20]. Based on these results, the amounts of antileishmanial flavonoids ingested by flies could vary substantially in different landscapes.

Besides floral nectar, other known sugar sources may also possess flavonoid-mediated antileishmanial activity. Flavonoid concentrations of sand fly-attracting fruits [26] appear similar to those found in nectar. Combined quercetin and luteolin contents ranged from unquantifiable in honeydew melon to 22.8 μM in nectarine, 33.1 and 39.7 μM in red and white guava, and 53.6, 84.1, and 91.1 μM in white, black, and red grapes respectively [62]. Honeydew from sapsucking insects such as aphids [12] likely also contains types and quantities of flavonoids similar to those found in floral nectar, based on the similar flavonoid profiles of honey from these two sources [63]. Further experiments are needed to assess the chemistry of local, fly-attracting, sugar-providing plant species and their effects on insect host and parasite mortality, as demonstrated for lectin-rich plant sugar sources in Israel [14,15,22]. Given that sand fly feeding on branches [19], flowers [12], and fruits [26] tends to be highly selective on a few local species, the scope of such research is likely achievable.

We predict that our analysis—which accounts only for direct effects of a few compounds on parasites as estimated from *in vitro* studies—provides a conservative estimate of the effects of plant compounds on disease transmission. First, we focused on a limited subset of nectar components whose effects on *Leishmania* have been thoroughly studied, ignoring the effects of co-occurring chemicals that could also affect parasites (e.g., other flavonoids, lectins, and alkaloids), whether present in the ingested sugar source or formed during sand fly digestion (e.g., deglycosylation of coumarins or cyanogenic glycosides to compounds that reduce parasitic infection [16,64]). Second, these direct effects could be amplified by host-mediated reductions in levels of parasites due to phytochemical ingestion. For example, nectar-derived flavonoids stimulated immune gene expression in honey bees [65]; similar flavonoid-induced immune stimulation could enhance parasite clearance in flies. In addition, besides their effects on protozoa specifically, flavonoids are generally antimicrobial [38], and could inhibit growth of midgut bacteria that facilitate *Leishmania* infection [66]. It would be of interest to contrast the effects of similar flavonoid concentrations taken directly from plant tissues—which are delivered to the sand fly midgut—versus those from surface sugars (e.g., nectar and honeydew),

which are first stored in the crop [57]. The gradual release of nectar and honeydew from the crop to into the anterior midgut [57] could limit the exposure of midgut-dwelling parasites to phytochemicals from these sources.

The effect of sugar-containing meals from plant sources was a long-overlooked component of sand fly ecology that proved crucial in *Leishmania* transmission [17]. However, feeding of sand flies on several plant taxa causes marked parasite mortality—up to 88% in the case of castor bean (*Ricinus communis*) [14], the lectins of which agglutinate a variety of insect trypanosomatids [67]—paralleling the strong effects of plant sugar sources on malaria infection in mosquitoes [13]. Although flowering plant nectar sources might at first glance appear to be a liability for *Leishmania* transmission due to the food they provide to sand flies, sugar starvation in fact results in greater vector infection intensity and natural selection for flies with lesser parasite resistance [68,69]. This finding is consistent with the preponderance of *Leishmania* hotspots in arid regions, where plant sugar sources are scarce [68,69]. High parasite loads also alter sand fly feeding on mammals in ways that promote transmission to new hosts [70]. These lines of evidence suggest that despite their role as vector food sources, phytochemical-rich floral nectar sources could have a net transmission-reducing effect.

Feeding of sand flies on floral nectar may also result in incidental pollen exposure that, due to pollen's high flavonoid concentrations, has strong effects on *Leishmania* in the fly gut. Such incidental exposure was suggested by the high prevalence of Pinaceae DNA associated with sand flies at sites apparently lacking such plants [20]. This association was postulated to reflect exposure of flies to windblown pollen, which could also account for at least some of the DNA from *Cannabis sativa*—another wind-pollinated species not visibly present [20]. Introduction of pollen to nectaries by bees can increase nectar amino acid concentrations by an order of magnitude, and potentially introduce antiparasitic compounds from con- and heterospecific pollens as well [50]. Given that flavonoid concentrations in pollen are 200-fold higher than those in nectar [40], incidental ingestion of even small amounts of pollen could substantively inhibit proliferation of parasites and the transmission potential of their vectors. In *H. annuus*, pollen-associated spermidines occurred at concentrations >1% of those in pollen even when pollinators were excluded [40]. In our meta-analysis, eight of the 28 pollens previously surveyed contained flavonoids at concentrations that exceeded 100-fold the maximum inhibitory concentration reported for *Leishmania* (Fig 2). This suggests that as little as 1% incidental addition of pollen to nectaries might be sufficient for *Leishmania* inhibition, even for nectars that lack antileishmanial flavonoids initially.

Whether antiparasitic compounds are present in secreted nectars or due to incidental introduction of pollen, nectars rich in phytochemicals are promising candidates for ecological mitigation of *Leishmania* transmission. Parasites of this genus appear both sensitive to flavonoids and, given the parasite's establishment in the midgut and forward migration in the alimentary canal [17], directly exposed to ingested compounds before appreciable metabolism of these compounds—by hosts or microbiota in the abdominal midgut—can occur. The limited intestinal absorption of ingested flavonoids [49], hydrolysis of glycosides found in plants to their more potent aglycones in the intestine [28,49], and likelihood of direct contact between parasites in the anterior midgut and ingested phytochemicals all indicate the potential for flavonoid-rich nectars to reduce *Leishmania* infection in sand flies. However, empirical testing of these compounds in sand fly diets is necessary to confirm their efficacy in the insect vector and model the effects of sugar sources on parasite infection, vector longevity, and disease transmission, as was recently done for malaria [13]. In addition, the broader ecological effects of floral compounds—whose effects are unlikely to be limited to sand flies alone—must be considered before implementation of interventions, particularly those that involve introduction of non-endemic plant species.

## Conclusions

The global toll of *Leishmania* infection and the difficulties of eradicating its sand fly vectors and non-human reservoirs demand the development of new, environmentally compatible strategies to reduce parasite transmission [71]. Our synthesis of existing data shows that sugar-seeking sand flies are attracted to floral resources, and that floral nectars contain antileishmanial phytochemicals at concentrations that inhibit replication of parasite cell cultures. The extent to which floral resources influence *Leishmania* epidemiology will depend on the contribution of nectar to sand fly diets and the extent to which *in vitro* inhibitory effects are realized in the guts of infected flies. If the effects of nectar on insect infection are commensurate with predictions based on nectar phytochemistry, reduction of transmission *via* supply of antiparasitic nectar sources in local landscapes—or phytochemical-based, transmission-blocking baits [16]–could positively influence public health. Such interventions could reduce reliance on drug treatments that may be costly, inaccessible, or potentially hazardous [3] while simultaneously supporting populations of beneficial insects and their resistance to insect-specific trypanosomatid infections [29,30]. The fields of insect ecology and medicinal chemistry for insect-vectored parasites have thus far developed more in parallel than in concert. Integrating knowledge of medicinal plant chemistry and plant-mediated tritrophic interactions that affect parasites in disease-vectoring insects holds promise for environmentally friendly control of trypanosomatid threats to global health.

## Supporting information

**S1 Text. Supplementary Table A, references, and metadata.**
(PDF)

**S1 Data. Zipped folder with data spreadsheets for *Leishmania* inhibitory concentrations (leishmania_ic50) and nectar and pollen flavonoid concentrations (nectar.pollen.flavonoids).**
(ZIP)

## Acknowledgments

We thank the many researchers whose work made this synthetic analysis possible.

## Author Contributions

**Conceptualization:** Evan C. Palmer-Young.

**Data curation:** Evan C. Palmer-Young.

**Formal analysis:** Evan C. Palmer-Young.

**Funding acquisition:** Ryan S. Schwarz, Yanping Chen.

**Investigation:** Evan C. Palmer-Young, Ryan S. Schwarz.

**Methodology:** Evan C. Palmer-Young, Ryan S. Schwarz.

**Resources:** Ryan S. Schwarz, Jay D. Evans.

**Supervision:** Jay D. Evans.

**Visualization:** Evan C. Palmer-Young.

**Writing – original draft:** Evan C. Palmer-Young.

**Writing – review & editing:** Evan C. Palmer-Young, Ryan S. Schwarz, Yanping Chen, Jay D. Evans.

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
