## [Decision Letter · Decision Letter 0]

6 Dec 2021

Dear Mr. Palmer-Young,

Thank you very much for submitting your manuscript "Can floral nectars reduce transmission of Leishmania?" for consideration at PLOS Neglected Tropical Diseases. As with all papers reviewed by the journal, your manuscript was reviewed by members of the editorial board and by several independent reviewers. In light of the reviews (below this email), we would like to invite the resubmission of a significantly-revised version that takes into account the reviewers' comments. 

Please, verify the comments of the reviewers carefully.

We cannot make any decision about publication until we have seen the revised manuscript and your response to the reviewers' comments. Your revised manuscript is also likely to be sent to reviewers for further evaluation.

Sincerely,

Claudia Ida Brodskyn

Associate Editor

Alvaro Acosta-Serrano

Deputy Editor

Please, verify the comments of the reviewers carefully.

Reviewer's Responses to Questions

**Key Review Criteria Required for Acceptance?**

**Methods**

-Are the objectives of the study clearly articulated with a clear testable hypothesis stated?

-Is the study design appropriate to address the stated objectives?

-Is the population clearly described and appropriate for the hypothesis being tested?

-Is the sample size sufficient to ensure adequate power to address the hypothesis being tested?

-Were correct statistical analysis used to support conclusions?

-Are there concerns about ethical or regulatory requirements being met?

Reviewer #1: As this is a meta-analysis their is no testable hypothesis. So the other questions are irrelevant

Reviewer #2: This section methods is fine for me except some points to address. Please find below my comments

1- Line 101-102 (Ref 27), could you explain how flavonoid glycosides are less effective than aglycones?

Reviewer #3: The study design was clear, however it is not strong enough support the hypothesis raised in this study.

**Results**

-Does the analysis presented match the analysis plan?

-Are the results clearly and completely presented?

-Are the figures (Tables, Images) of sufficient quality for clarity?

Reviewer #1: As this is a compilation of other scientists' research there was no analysis plan to begin with. 

The results are clearly stated.

Figures:

Figure 1 A: There are no values even close to 1,000 (1e+03), so why have a graph that goes to 1e+05 (100,000)? B: Is very busy and hard to read – can it be simplified? 

Figures 2 & 3. N.B. fig. 2 uses 1e + scale, while fig 3 uses 0.1 to 100. The second is so much easier to read. 

Table 

Table includes data on amastigotes, a stage in the life cycle of the parasite that is very transitory in the sand fly.

Reviewer #2: Good, but is the concentration of floral resources that you use the same concentration that could Phlobotomus ingest in the wild?

Reviewer #3: Concentrations evaluated here are based only in vitro exposition of the parasite to compounds, ignoring their effect during Leishmania-sand fly interactions, i.e., ignoring the behavior and physiology of the insect host, in which is essential to the potential for the vector control.

**Conclusions**

-Are the conclusions supported by the data presented?

-Are the limitations of analysis clearly described?

-Do the authors discuss how these data can be helpful to advance our understanding of the topic under study?

-Is public health relevance addressed?

Reviewer #1: The conclusions are adequate, but the limitations are (i.e. lack of field data) not mentioned.

No evidence is presented that sand flies actually feed on the various plants mentioned.

In the field of Public Health the authors wrote that their work could lead to an environmentally friendly control of trypanosomatid threats to global health.

Reviewer #2: Good

Reviewer #3: The compilation done here would be extremely helpful as a screening process to test experimentally these compounds in the insect vector. Such conclusion would be more restrictive, but certainly more accurate to the data presented.

**Editorial and Data Presentation Modifications?**

Reviewer #1: Lines 13 and 53: Insect-vectored Leishmania are the second-most debilitating of human parasites worldwide. The actual quote is “Among parasitic infections, this disease is responsible for the highest number of disability adjusted life years (a measure of health burden) after malaria.” (Lymphatic Filariasis is surely more debilitating than cutaneous leishmaniasis caused by Leishmania major). While the authors repeat several times this health burden, it is somewhat over empathized. 

Line 54 The CDC numbers: For cutaneous leishmaniasis, estimates of the number of new cases per year have ranged from approximately 700,000 to 1.2 million or more. For visceral leishmaniasis, the estimated number of new cases per year may have decreased to <100,000. WHO numbers are less, so perhaps “reported” should be cautiously used.

Line 66: “floral nectar appears to be a preferred food source” Perhaps it should be noted that this study was carried out in a lush irrigated farming village surrounded by desert.

Line 137 and Discussion: Is there any evidence that sand flies feed actually on for example (Dicentra eximia, Brassica napus, Helianthus annuus, or Thymus vulgaris)? Examples of mesocosm would augment the meta-analysis. A real world example of this would be “DNA barcode for the identification of the sand fly Lutzomyia longipalpis plant feeding preferences in a tropical urban environment”: Leonardo H G de M Lima et al Sci Rep, 2016 Jul 20;6: 29742 or if the authors comb the literature they can find Suaeda asphaltica as a sand fly food source or Cameron’s work in South America on bean plants as actual sand fly food or Alexander B and Usma MC, study in Colombian coffee fields.

Could it be that the compounds the authors are promoting would make better pharmaceutical products? 

While the MS is partly speculative in nature, the inclusion of some of the references add little to overall question. 

Other references that may interest the authors for their discussion:

Lectins and toxins in the plant diet of Phlebotomus papatasi (Diptera: Psychodidae) can kill Leishmania major promastigotes in the sandfly and in culture. Jacobson RL, Schlein Y. Ann Trop Med Parasitol. 1999 Jun;93 (4):351-6.

Sand fly feeding on noxious plants: a potential method for the control of leishmaniasis Y. Schlein, R L Jacobson, G C Müller Am J Trop Med Hyg. 2001 Oct; 65(4):300-3

While the MS is partly speculative in nature, the inclusion of some of the references adds little to overall question?

Reviewer #2: Good

Reviewer #3: (No Response)

**Summary and General Comments**

Reviewer #1: The MS reviews the research of others on the in vitro screening of various extracts of nectar and pollen on Leishmania species and discuss the use of secondary metabolites in the control of the infections in the sand fly vector.

The inclusion of known plant food sources would improve the MS

Reviewer #2: Phlebotomus take the nectar as nutrition. The effects of components in nectar are various. This study focuses on floral nectars and Leishmania transmission. The conclusion is that planting phytochemical-rich floral resources or phytochemically enriched baits could reduce Leishmania loads in vectors, providing an environmentally friendly complement to existing means of disease control. With these findings, there is a great hope to come over some diseases as Leishmaniasis without chemical solution, using natural environmental resources.

The manuscript how it is standing now, is well written, clear and shows good results to fight against the Leishmania. However, to improve the manuscript some points raised below need to be more detailed. 

1- Line 117-118 (Ref 13), in this paper they have shown that some plant species reduce the transmission of malaria and other plants maintain the transmission, so, could you consider this in the discussion?

2- In your references 2, 12, 17, 23 there is no DOI as well as in all your references in Supporting information 1, please add these details

3- Some species scientific names in your references are not italicized, you should correct that too

Reviewer #3: The information discussed in this study is certainly interesting, but it is not being consistent with the broader potential speculated by authors. I suggest authors either support the study with some experimental tests of nectar compounds on insects or restrict claims to avoid the speculative character.

PLOS authors have the option to publish the peer review history of their article (what does this mean?). If published, this will include your full peer review and any attached files.

Reviewer #1: No

Reviewer #2: No

Reviewer #3: No
---

## [Decision Letter · Decision Letter 1]

11 Feb 2022

Dear Palmer-Young,

Thank you very much for submitting your manuscript "Can floral nectars reduce transmission of Leishmania?" for consideration at PLOS Neglected Tropical Diseases. As with all papers reviewed by the journal, your manuscript was reviewed by members of the editorial board and by several independent reviewers. In light of the reviews (below this email), we would like to invite the resubmission of a significantly-revised version that takes into account the reviewers' comments. 

Although the modifications in the text have improved the manuscript, there are still some alterations that deserve attention by the authors.

We cannot make any decision about publication until we have seen the revised manuscript and your response to the reviewers' comments. Your revised manuscript is also likely to be sent to reviewers for further evaluation.

Sincerely,

Claudia Ida Brodskyn

Associate Editor

Alvaro Acosta-Serrano

Deputy Editor

Although the modifications inthe text have improved the manuscript, there are still some alterations that deserve attention by the authors.

Reviewer's Responses to Questions

**Key Review Criteria Required for Acceptance?**

**Methods**

-Are the objectives of the study clearly articulated with a clear testable hypothesis stated?

-Is the study design appropriate to address the stated objectives?

-Is the population clearly described and appropriate for the hypothesis being tested?

-Is the sample size sufficient to ensure adequate power to address the hypothesis being tested?

-Were correct statistical analysis used to support conclusions?

-Are there concerns about ethical or regulatory requirements being met?

Reviewer #1: (No Response)

Reviewer #2: Good, very clear to me now

Reviewer #3: As previously mentioned the study is clear and presents a satisfactory design.

**Results**

-Does the analysis presented match the analysis plan?

-Are the results clearly and completely presented?

-Are the figures (Tables, Images) of sufficient quality for clarity?

Reviewer #1: (No Response)

Reviewer #2: Good

Reviewer #3: Data presented previously was clear and modifications have not big effect in the current manuscript.

**Conclusions**

-Are the conclusions supported by the data presented?

-Are the limitations of analysis clearly described?

-Do the authors discuss how these data can be helpful to advance our understanding of the topic under study?

-Is public health relevance addressed?

Reviewer #1: (No Response)

Reviewer #2: Good

Reviewer #3: In my view, the conclusion of this work still remain very broad considering the data based only in vitro tests to discuss a bigger question, a more restrictive statement would be more plausible.

**Editorial and Data Presentation Modifications?**

Reviewer #1: (No Response)

Reviewer #2: Good

Reviewer #3: I acknowledge all changes performed by the authors in the text, it improved considerably the manuscript, however I still would recommend some "major revision" in this work.

**Summary and General Comments**

Reviewer #1: The authors have used the remarks of the referees to refine their MS. It has been improved.

Reviewer #2: Good from my point of view

Reviewer #3: Despite all modifications performed, the hypothesis raised in this work still consider a large knowledge in nectar compounds and their effect in the parasite in vitro, it has poor information related to the vector interaction, in which is essential considering to the potential claimed in this manuscript.

 I still recommend authors discuss more about the physiology and behaviour of the insect that could justify the hypothesis. Even simple details regarding the insect could make all the difference and should be not selectively ignored. For instance, if the nectar amount taken by the insect would enough to affect the parasite considering the in vitro results presented here? 

Please check here some papers that might help :

Ferreira, T.N., Pita-Pereira, D., Costa, S.G. et al. Transmission blocking sugar baits for the control of Leishmania development inside sand flies using environmentally friendly beta-glycosides and their aglycones. Parasites Vectors 11, 614 (2018). https://doi.org/10.1186/s13071-018-3122-z

Ferreira ME, de Arias AR, Yaluff G, de Bilbao NV, Nakayama H, Torres S, et al. Antileishmanial activity of furoquinolines and coumarins from Helietta apiculata. Phytomedicine. 2010;17:375–8.

Gontijo NF, Melo MN, Riani EB, Aleida-Silva S, Mares-Guia ML. Glycosidases in Leishmania and their importance for Leishmania in phlebotomine sand flies with special reference to purification and characterization of a sucrase. Exp Parasitol. 1996;83:117–24.

Jacobson RL, Schlein Y, Eisenberger CL. The biological function of sand fly and Leishmania glycosidases. Med Microbiol Immunol. 2001;190:51–5

Dillon RJ, EEl K. Carbohydrate digestion in sand flies: α-glucosidase activity in the midgut of Phlebotomus langeroni. Comp Biochem Physiol. 1997;116B:35–40.

PLOS authors have the option to publish the peer review history of their article (what does this mean?). If published, this will include your full peer review and any attached files.

Reviewer #1: No

Reviewer #2: No

Reviewer #3: No
---

## [Decision Letter · Decision Letter 2]

9 Mar 2022

Dear Palmer-Young,

Thank you very much for submitting your manuscript "Can floral nectars reduce transmission of Leishmania?" for consideration at PLOS Neglected Tropical Diseases. As with all papers reviewed by the journal, your manuscript was reviewed by members of the editorial board and by several independent reviewers. The reviewers appreciated the attention to an important topic. Based on the reviews, we are likely to accept this manuscript for publication, providing that you modify the manuscript according to the review recommendations. 

Dear authors,

The manuscript is improved a lot, but two reviewers pointed out some minor issues that deserve your attention.

Please, answer these points and send the manuscript back to be analyzed again.

Sincerely,

Claudia Ida Brodskyn

Associate Editor

Alvaro Acosta-Serrano

Deputy Editor

Dear all,

The manuscript is improved a lot, but two reviewers pointed out some issues that deserve your attention.

Please, answer these points and send the manuscript back to be analyzed agains.

Reviewer's Responses to Questions

**Key Review Criteria Required for Acceptance?**

**Methods**

-Are the objectives of the study clearly articulated with a clear testable hypothesis stated?

-Is the study design appropriate to address the stated objectives?

-Is the population clearly described and appropriate for the hypothesis being tested?

-Is the sample size sufficient to ensure adequate power to address the hypothesis being tested?

-Were correct statistical analysis used to support conclusions?

-Are there concerns about ethical or regulatory requirements being met?

Reviewer #2: Good for me

Reviewer #3: Methods are clear and well described by the authors

Reviewer #4: (No Response)

**Results**

-Does the analysis presented match the analysis plan?

-Are the results clearly and completely presented?

-Are the figures (Tables, Images) of sufficient quality for clarity?

Reviewer #2: Good

Reviewer #3: Result analysis is consistent with the study design and clearly explained.

Reviewer #4: see below

**Conclusions**

-Are the conclusions supported by the data presented?

-Are the limitations of analysis clearly described?

-Do the authors discuss how these data can be helpful to advance our understanding of the topic under study?

-Is public health relevance addressed?

Reviewer #2: Good

Reviewer #3: Lines: 289-292

I would like just to point that some technical interspecies issues may need to be addressed here when extrapolating these results. It is unlikely that the phytochemistry from these floral compounds would be specific to sand flies. The ecological concerns would be similar to using synthetic compounds.

Reviewer #4: see below

**Editorial and Data Presentation Modifications?**

Reviewer #2: Good

Reviewer #3: Accept

Reviewer #4: see below

**Summary and General Comments**

Reviewer #2: Good for me now, but they need to consider changing Phlebotomus langeroni in italic in reference 23 and the figure 1 in supporting information section is blurred

Reviewer #3: I acknowledge all modifications performed in the manuscript. The text is more consistent with the data analysed.

Reviewer #4: Revised version of the manuscript is significantly improved as helpful comments of reviewers were accepted. In my opinion, just few relatively minor changes are required:

Authors are aware that the sugar meal is directed first to crop (see lines 237-240). However, I am missing important information that only small droplets of sugar meal are released from the crop to the anterior (thoracic) midgut where the sugar meal is relatively quickly digested by glycosidases. This mechanism of sugar digestion may significantly decrease the effect of anti-leishmania compounds in the sugar meal. It is necessary to mention this fact either in Discussion or Conclusions.

The above mentioned fact should be also reflected by more cautious deductions in the text, for example „would be“ should be replaced by „might be“ on line 267. 

Line 274: the proportion of microbiota in sand fly hindgut is negligible and unimportant in comparison to microbiota present in abdominal midgut of sand flies. Therefore, „hindgut microbiota“ should be replaced by „microbiota in abdominal midgut“.

PLOS authors have the option to publish the peer review history of their article (what does this mean?). If published, this will include your full peer review and any attached files.

Reviewer #2: No

Reviewer #3: No

Reviewer #4: No

Figure Files:

Data Requirements:

Reproducibility:

References

---

## [Editor Report · Decision Letter 3]

29 Mar 2022

Dear Palmer-Young,

We are pleased to inform you that your manuscript 'Can floral nectars reduce transmission of Leishmania?' has been provisionally accepted for publication in PLOS Neglected Tropical Diseases.

Best regards,

Claudia Ida Brodskyn

Associate Editor

Alvaro Acosta-Serrano

Deputy Editor

After three rounds of revision, the manuscript can be accepted for publication.\\Best regards

Cláudia Brodskyn

---

## [Editor Report · Acceptance letter]

16 Apr 2022

Dear Palmer-Young,

We are delighted to inform you that your manuscript, "Can floral nectars reduce transmission of Leishmania?," has been formally accepted for publication in PLOS Neglected Tropical Diseases.

Best regards,

Shaden Kamhawi

co-Editor-in-Chief

Paul Brindley

co-Editor-in-Chief
